# Six-Month Results on Treatment Adherence, Physical Activity, Spinal Appearance, Spinal Deformity, and Quality of Life in an Ongoing Randomised Trial on Conservative Treatment for Adolescent Idiopathic Scoliosis (CONTRAIS)

**DOI:** 10.3390/jcm10214967

**Published:** 2021-10-26

**Authors:** Marlene Dufvenberg, Elias Diarbakerli, Anastasios Charalampidis, Birgitta Öberg, Hans Tropp, Anna Aspberg Ahl, Hans Möller, Paul Gerdhem, Allan Abbott

**Affiliations:** 1Department of Health, Medicine and Caring Sciences, Unit of Physiotherapy, Linköping University, SE 581 83 Linköping, Sweden; birgitta.oberg@liu.se (B.Ö.); allan.abbott@liu.se (A.A.); 2Department of Clinical Science, Intervention and Technology (CLINTEC), Division of Orthopaedics and Biotechnology, Karolinska Institutet, SE 141 86 Stockholm, Sweden; elias.diarbakerli@sll.se (E.D.); anastasios.charalampidis@sll.se (A.C.); hans.moller@rkc.se (H.M.); paul.gerdhem@sll.se (P.G.); 3Department of Reconstructive Orthopaedics, Karolinska University Hospital Huddinge, SE 141 86 Stockholm, Sweden; 4Department of Biomedical and Clinical Sciences, Linköping University, SE 581 83 Linköping, Sweden; hans.tropp@regionostergotland.se; 5Center for Medical Image Science and Visualization, Linköping University, SE 581 83 Linköping, Sweden; 6Department of Orthopaedics, Linköping University Hospital, SE 581 83 Linköping, Sweden; 7Department of Orthopaedics, Ryhov County Hospital, SE 551 85 Jönköping, Sweden; anna.aspberg.ahl@rjl.se; 8Stockholm Center for Spine Surgery, SE 171 64 Stockholm, Sweden

**Keywords:** idiopathic scoliosis, bracing, physiotherapeutic scoliosis-specific exercise, physical activity, adherence, spinal appearance, health-related quality of life

## Abstract

Adolescents with idiopathic scoliosis (AIS) often receive conservative treatments aiming to prevent progression of the spinal deformity during puberty. This study aimed to explore patient adherence and secondary outcomes during the first 6 months in an ongoing randomised controlled trial of three treatment interventions. Interventions consisted of physical activity combined with either hypercorrective Boston brace night shift (NB), scoliosis-specific exercise (SSE), or physical activity alone (PA). Measures at baseline and 6 months included angle of trunk rotation (ATR), Cobb angle, International Physical Activity Questionnaire short form (IPAQ-SF), pictorial Spinal Appearance Questionnaire (pSAQ), Scoliosis Research Society (SRS-22r), EuroQol 5-Dimensions Youth (EQ-5D-Y) and Visual Analogue Scale (EQ-VAS). Patient adherence, motivation, and capability in performing the intervention were reported at 6 months. The study included 135 patients (111 females) with AIS and >1-year estimated remaining growth, mean age 12.7 (1.4) years, and mean Cobb angle 31 (±5.3). At 6 months, the proportion of patients in the groups reporting high to very high adherence ranged between 72 and 95%, while motivation ranged between 65 and 92%, with the highest proportion seen in the NB group (*p* = 0.014, *p*= 0.002). IPAQ-SF displayed significant between group main effects regarding moderate activity (F = 5.7; *p* = 0.004; η_p^2^_ = 0.10), with a medium-sized increase favouring the SSE group compared to NB. Walking showed significant between group main effects, as did metabolic equivalent (MET-min/week), with medium (F = 6.8, *p* = 0.002; η_p^2^_ = 0.11, and large (F = 8.3, *p* = < 0.001, η_p^2^_ = 0.14) increases, respectively, for the SSE and PA groups compared to NB. From baseline to 6 months, ATR showed significant between group medium-sized main effects (F = 1.2, *p* = 0.019, η_p^2^_ = 0.007) favouring the NB group compared to PA, but not reaching a clinically relevant level. In conclusion, patients reported high adherence and motivation to treatment, especially in the NB group. Patients in the SSE and PA groups increased their physical activity levels without other clinically relevant differences between groups in other clinical measures or patient-reported outcomes. The results suggest that the prescribed treatments are viable first-step options during the first 6 months.

## 1. Introduction

Idiopathic scoliosis is a deformity of the spine, with most cases developing during puberty [1,2]. Spinal deformity can negatively affect health-related quality of life (HRQoL), including psychological distress, perception of spinal appearance, back pain, and pulmonary function in large curves [3,4,5,6,7]. Conservative treatments including a combination of bracing, scoliosis-specific exercise, and more general physical activity interventions are commonly used for patients with moderate-grade AIS, in order to prevent progression and to avoid surgical treatment [8].

Thoracolumbosacral braces are designed to provide three-dimensional correctional counterforces to the scoliosis curvature. Full-time bracing has been the usual care intervention for moderate-grade AIS [8]. A systematic review provides low-quality strength of evidence suggesting that full-time bracing is effective in approximately 75% of cases in preventing excessive progression of the spinal deformity during puberty [9]. More hours of brace wear have been associated with higher success rates [10]. However, full-time bracing can negatively affect patients’ perceptions of body appearance, reduce quality of life, and cause back pain [11,12]. Hypercorrective night-time bracing aims to provide curve correction in supine position with reduced mechanical loading [13], and has shown comparable outcomes to full-time bracing in halting the progress of spine deformity [14,15]. Night-time bracing may also minimise potential limitation of daytime activities compared to full-time bracing, and may facilitate better treatment acceptability and adherence for a larger proportion of patients. This warrants investigating the viability of night-time bracing as a first-step approach before considering full-time bracing. This may potentially mitigate negative side effects and overtreatment of bracing [10,16,17].

Current international guidelines recommend scoliosis-specific exercise (SSE) to treat mild AIS (Cobb angle 11–24°), or adjunct to bracing for moderate-grade AIS [8]. However, it is unknown whether SSE is a viable first-step treatment for moderate-grade AIS. Standard features of SSE include three-dimensional spinal self-correction strategies and their integration in activities of daily living, stabilizing the corrected posture, and patient education [8,18]. This potentially influences neuromuscular and sensory integration effect mechanisms, aiming to prevent progression of AIS [19]. A recent systematic review comparing SSE with other conservative interventions for AIS displayed low-quality evidence for SSE causing greater improvements in function, HRQoL, self-image, mental health, and patient satisfaction. However, bracing displayed more effectiveness on measures of spinal deformity than SSE [20].

Physical activity is associated with health benefits for children, including improved muscle strength, motor development, aerobic fitness, and higher bone mineral density [21]. Children are recommended to perform a daily minimum of 60 min of physical activity of at least moderate intensity [21,22]. Current AIS guidelines also recommend sporting activities [8]. According to studies proposing low bone mineral density as a possible mechanism behind AIS [23,24], physical activity might be a possible alternative intervention [25]. However, research is needed in order to evaluate whether physical-activity-related intervention alone is also viable in preventing the progression of moderate-grade AIS [8].

Adherence can be defined as to what extent a person’s behaviour corresponds to treatment recommendations [26]. Poor adherence is a reason for suboptimal clinical effect [26], and is therefore important to monitor in clinical trials where the treatment is self-managed and requires long-term maintenance.

An ongoing randomised controlled trial (RCT)—Conservative Treatment for Adolescent Idiopathic Scoliosis (CONTRAIS)—aims to investigate the effectiveness of conservative treatments of moderate-grade AIS in preventing progression and the need for surgical intervention [27]. The treatments consist of adequate self-mediated physical activity levels combined with either hypercorrective Boston brace night shift (NB), scoliosis-specific exercise (SSE), or active control with adequate self-mediated physical activity alone (PA).

Beyond assessment of Cobb angle and trunk rotation, it is of importance in the short-term to explore patients’ adherence to treatments, physical activity levels, and potential effects on perception of spinal appearance and HRQoL, considering the long-term maintenance required for scoliosis treatments [28]. The aim of the present study was to explore patient adherence to treatment and secondary outcomes during the first 6 months in terms of physical activity, spinal appearance, spinal deformity, and quality of life within and between NB-, SSE-, and PA-based treatments for AIS in an ongoing RCT.

## 2. Materials and Methods

### 2.1. Study Design and Selection of Participants

The present study reports data at baseline and 6 months of an ongoing RCT. The study protocol has been previously published, and is available on clinicaltrials.gov (NCT0176130) [27]. Approval has been obtained by the Regional Ethical Board in Stockholm (Dnr 2012/172-31/4, 2015/1007-32, and 2017/609-32). Study centres were the orthopaedic departments at Karolinska University Hospital, Linköping University Hospital, Ryhov Hospital Jönköping, Eskilstuna Hospital, Västerås Hospital, and Sundsvall Hospital.

Inclusion criteria encompassed a diagnosis of idiopathic scoliosis in females and males; primary curve according to Cobb of 25–40° [29]; curve apex T7 or caudal; estimated remaining skeletal growth of at least one year; not more than one year after menarche; and age 9–17. Exclusion criteria were previous treatment for scoliosis or inability to understand Swedish. The enrolment of the patients took place consecutively between January 2013 and October 2018. Patients who declined participation were offered standard care, which consisted of full-time bracing. In total, 135 patients were randomised into one of three treatment groups: NB, SSE, or PA (Figure 1).

The present study concerns secondary outcomes based on clinical measurements and questionnaires at baseline and after 6 months of treatment. Questionnaires were completed without guidance of the health care provider (HCP), but guidance from the parents was allowed if requested by the patient. In the ongoing RCT, the primary outcome is failure of treatment, defined as an increase in Cobb angle of more than 6 degrees, as seen on two consecutive X-rays after 6 months in relation to the baseline X-ray. Our primary outcome will be reported in a future publication when all patients have reached the endpoint. The endpoint is defined as when the participant reaches skeletal maturity (less than 1.0 cm growth of body height in 6 months), or if the curve progresses by more than 6 degrees, as seen on two consecutive X-rays after 6 months in relation to the baseline X-ray.

A computer-generated concealed randomisation with varying block sizes, at a 1:1:1 ratio, was prepared a priori by an independent statistician. After informed consent, a central research coordinator was contacted for the randomisation process, after which the participants were assigned to the interventions. In this study, blinding of patients and therapists for treatment was not possible.

### 2.2. Interventions

To enhance desired behavioural outcomes—i.e., high adherence to the treatment plan and self-management—the capability, opportunity, motivation, and behaviour change model (COM-B) was applied for all allocated treatments [30]. The HCPs gave instructions and guidance to enable the patients’ self-care ability, goal setting, progression, and follow-up of intervention prescription diary monitoring, as well as parents’ participation in the treatment delivery. At an initial 60 min individual session with one out of four experienced physiotherapists, patients were instructed to increase physical activity for the entirety of the study. Instructions were given to perform adequate self-mediated physical activity of at least moderate intensity for ≥ 60 min daily in all groups [31]. A summary of the additional interventions is provided below, and in more detail in Appendix A. Abbreviations in Appendix A. Every 6 months, reinforcement of the assigned intervention was performed Appendix A at the clinic during an individual session of 60 min. Additional contact via telephone was used when needed.

#### 2.2.1. Hypercorrective Boston Brace Night Shift

The NB group received a hypercorrective Boston brace night shift [32] tailored to the individual patients’ scoliosis. and applying built-in correction in all three planes. The criterion for adequate brace correction was greater than or equal to 50% reduction in Cobb angle. Patients were advised to use the brace throughout every night for at least 8 h. The physiotherapist and orthotist educated the patient and parent when introducing the brace during the first session. The spine orthotist was available for outpatient brace adjustment when needed.

#### 2.2.2. Scoliosis-Specific Exercise

The SSE group received supervised individual treatment by one out of four experienced physiotherapists, consisting of 3 × 60–90 min individual sessions with reiteration of skills learned and skill progression, once per month for the first 3 months. Additional single bolus sessions were given if required. Training goals were directed towards active self-correction in 3 dimensional planes, postural control, spinal stability, muscular stabilisation, and endurance in corrective postures, and were integrated into activities of daily living. Patients were initially asked to perform these specific exercises with 10 repetitions × 3 sets over 30 min, which was included in their prescription of 60 min self-mediated physical activity. When patients had mastered these exercises, they could additionally focus on transferring these skills to similar activities of daily living or individual sporting and recreational activities of interest. Patients were also instructed to use overcorrective side-shift postural strategies during activities of daily living involving relaxed sitting and standing positions. The SSE covered similar concepts and methods to those described in previous literature [33]. The physiotherapists were trained to ensure consistent treatment approach.

#### 2.2.3. Active Control with Adequate Self-Mediated Physical Activity

The PA active control group were instructed to perform adequate self-mediated physical activity of at least moderate intensity for ≥60 min daily, for the entirety of the study, with no additional bracing or SSE intervention [31].

### 2.3. Measurements

#### Patient Characteristics and Clinical Measures

Patient data such as age, gender, weight, height, and angle of trunk rotation (ATR) assessed via scoliometer were assessed at baseline and at 6 months. Adverse events were reported at 6 months. The assessments took place at the orthopaedic departments, and were coordinated by a multiprofessional team consisting of physicians, nurses, and physiotherapists. Whole standing spine radiographs were performed at baseline and after 6 months. Two experienced spine surgeons not otherwise part of the research group performed blinded assessment of Cobb angle measurements of the largest curve on anonymised images without knowledge of age, sex, or treatment. The mean of the two blinded assessors’ Cobb angle measurements of the largest curve was reported at baseline and 6 months.

### 2.4. Questionnaires, Self-Reported Data, and Health Care Provider Report

The questionnaires included self-report of maturity stages for breast/genitals and pubic hair (Tanner), treatment adherence, effects on HRQoL, perception of spinal appearance, and physical activity level.

#### 2.4.1. Adherence, Motivation, and Capability

In line with the COM-B model [30], to gain knowledge of desired behavioural outcomes—i.e., adherence, motivation, and capability regarding the treatment plan—all study participants answered three additional questions ”the grade to which you feel that you have completed the treatment (reported adherence)”, “the grade to which you are motivated to carry out the treatment (reported motivation)” and, finally, “how confident are you in your own capability to perform the treatment? (reported capability)”.

Additionally, the treating HCP team answered one question—“to what grade the patient has adhered to treatment plan” (reported patient adherence)—using the diary and after dialogue with the patients and families. The patients and HCPs were asked to rate each question on a scale from best “very sure” (1 point) to worst “not at all” (4 points). These questions were collected at the 6-month follow-up.

#### 2.4.2. Physical Activity

Self-reported levels of physical activity during the prior 7 days were captured using the International Physical Activity Questionnaire short form (IPAQ-SF). The IPAQ-SF has acceptable measurement properties [34,35], and provides the achieved mean in minutes of daily activity at the levels of “walking”, “moderate” and “vigorous” intensity activity, and “sitting”. The volume of activity is computed by weighting each type of activity by its energy requirements as metabolic equivalents (MET). The mean total score of MET min/week is provided by summation of the duration (in min) and frequency (days) [35,36]. The total score ranges from 0 min/day to a maximum of 180 min/day.

#### 2.4.3. Perception of Spinal Appearance

The pictorial part (pSAQ) [6] of the Spinal Appearance Questionnaire (SAQ) is based on the Walter Reed Visual Assessment Scale [37], and has shown good psychometric properties [38,39]. The patient’s perception of spinal shape asymmetry is based on 7 categories: body curve, rib prominence, flank prominence, head–chest–hips relationship, head position over hips, shoulder level, and spine prominence. Each category is graded on a scale from “no” (1 point) to “most severe” spinal deformity (5 points). The total score ranges from “least” (7 points) to “most” (35 points) distorted appearance.

#### 2.4.4. Health Related Quality of Life

The Scoliosis Research Society-22r (SRS-22r) is a valid and reliable scoliosis-specific patient-reported measure of HRQoL, and comprises 22 questions categorized into 5 domains (function, pain, self-image, mental health, and satisfaction) [40,41,42]. The domains, subscore (function, pain, self-image, mental health) and total score (all five domains) can range from “worst” (1 point) to “best” (5 points) [41]. The satisfaction domain and total score were included as measures after 6 months of treatment.

EuroQol 5-Dimensions Youth (EQ-5D-Y) is a valid and reliable generic questionnaire measuring HRQoL, where children from 8 years of age can report their own health [43,44,45,46,47,48]. The instrument comprises five dimensions: “mobility”, “looking after myself”, “doing usual activities”, “having pain or discomfort” and “feeling worried, sad, or unhappy”. Each dimension has three levels of severity: “no” (1 point), “some “(2 points) and “a lot of” (3 points) problems. A global assessment of current health state is measured with a visual analogue scale (EQ-VAS), from “worst” (0 points) to “best” (100 points) imaginable health state.

### 2.5. Statistical Methods and Analysis

This study’s original sample size calculation was based on the primary outcome measure, with a failure of 15% in each of the NB and SSE groups, and 45% in the PA group [49]. With a significance level of 5%, a power of 80%, and consideration for dropout of up to 20%, 45 patients were required in each of the treatment groups [27].

Data was double-entered, and any discrepancies were checked and corrected. Interrater reliability of Cobb angle measurements performed by two blinded assessors was calculated via interclass correlation coefficient (ICC 2,1). Descriptive statistics at baseline were presented as mean and standard deviation (±), or as number and proportions (no, %). Patient–HCP agreement in reporting of adherence was estimated with linearly weighted kappa (K), and the relative strength of agreement according to poor <0.00, slight 0.00–0.20, fair 0.21–0.40, moderate 0.41–0.60, substantial 0.61–0.80, and almost perfect 0.81–1.00 [50]. Logistic regression was used to assess the association of two independent factors (motivation and capability) with adherence, applying dichotomised ratings “a very high grade and high grade” and “low grade and not at all”. Statistics are presented as Nagelkerke pseudo R-squared (R^2^) and odds ratio (OR).

To analyse differences within groups, paired-sample t-tests were performed. Effect size was estimated with Cohen’s *d* as small 0.20, medium 0.50, and large 0.80 [51]. Between group analyses for continuous and categorical variables were performed using univariate analysis of variance (ANOVA) or chi-squared. For ANOVA, partial eta squared (η_p^2^_) was used as an estimated measure of effect size, where η_p^2^_ = 0.01 ~ small effect, η_p^2^_ = 0.06 ~ medium effect, and η_p^2^_ = 0.14 ~ large effect [51]. Two-sided statistical significance was considered if *p* < 0.05. Paretian analysis presents a classification of health change in EQ-5D-Y from baseline to 6 months towards worse, unchanged, better, or indeterminable [52]. To avoid overestimation for IPAQ physical activity levels, outliers were excluded by truncating to 180 max min/day and a maximum of 21 h of activity/week [35]. All analyses were performed per protocol using statistical analysis performed using the Statistical Package for the Social Sciences (SPSS) statistical software for Windows (SPSS V26, IBM Corporation, New York, NY, USA).

## 3. Results

### 3.1. Participant Flow and Baseline Characteristics

Participant flow throughout the study is displayed in Figure 1. In total, 2150 patients were seen as outpatients during the inclusion period, and 135 patients were included and randomly allocated to 45 patients per group. The mean age was 12.7 (±1.4), and 111 were females (82%). There were two female questionnaire non-responders at baseline—one allocated to the NB group and one to the SSE group. Treatment groups did not differ at baseline regarding patient characteristics, clinical measures, or patient-reported outcomes (Table 1, Table 2, Table 3, Table 4, Table 5 and Table 6).

**Table 1 jcm-10-04967-t001:** Baseline characteristics of the included patients.

	Overall Sample *n* = 135	NB *n* = 45	SSE *n* = 45	PA *n* = 45	*p*-Value
Age * (years)	12.7 (1.4)	12.7 (1.4)	12.6 (1.4)	12.6 (1.5)	0.892
Females (no.%)	111 (82)	39 (87)	33 (73)	39 (87)	0.161
Height * (cm)	158.1 (9.5)	157.2 (9.5)	158.1 (9.6)	159.0 (9.6)	0.678
Weight * (kg)	46.0 (9.2)	44.8 (9.3)	45.7 (9.0)	47.3 (9.4)	0.434
Body mass index *	18.3 (2.6)	18.0 (2.7)	18.2 (2.4)	18.6 (2.7)	0.504
Angle of trunk rotation * (degrees)	11.3 (3.1)	11.8 (2.7)	10.8 (3.3)	11.3 (3.1)	0.287
Tanner breast/genital N (%)					0.521
I	8 (6)	2 (5)	5 (12)	1 (2)	
II	23 (18)	8 (19)	5 (12)	10 (24)	
III	65 (52)	22 (52)	20 (49)	23 (55)	
IV	26 (21)	10 (24)	9 (22)	7 (17)	
V	3 (2)	-	2 (5)	1 (2)	
Tanner pubic hair N (%)					0.595
I	16 (13)	5 (12)	6 (15)	5 (12)	
II	19 (15)	6 (15)	6 (15)	7 (17)	
III	33 (27)	14 (34)	9 (22)	10 (24)	
IV	49 (40)	14 (34)	15 (38)	20 (48)	
V	6 (5)	2 (5)	4 (10)		
Cobb angle * (degrees)	31 (5.3)	32 (5.6)	31 (4.6)	31 (5.6)	0.433
Location of largest curve N (%)					0.970
Thoracic	99 (73)	32 (73)	33 (73)	33 (73)	
Thoracolumbar	21 (16)	7 (16)	6 (13)	8 (18)	
Lumbar	15 (11)	5 (11)	6 (13)	4 (9)	

* Values are given as mean and standard deviation (SD). N: number of patients; NB: hypercorrective Boston brace night shift; SSE: scoliosis-specific exercise; PA: adequate self-mediated physical activity; Tanner scale: breast/genitals and pubic hair I–V maturity stages; Cobb angle: largest curve of scoliosis measured with X-ray.

A total of 135 patients received the allocated treatment plan. At 6 months, there were 16 questionnaire non-responders (Figure 1), of whom 11 gave no reasons, 4 opted for a full-time brace, and 1 due to anxiety. This resulted at 6 months in 119 patients who provided patient-reported outcomes: 40 patients each in the NB and SSE groups, and 39 in the PA group. HCPs provided data collection regarding 132 patients: 44 in the NB group, 43 in the PA group, and 45 in the SSE group. At 6 months, patients or HCPs reported five adverse events in the NB group: sleeping problems during habituation period, awkwardness of staying overnight with friends, pressure towards ribs, redness and itchiness, and one unspecified. In the SSE group, one patient reported pain during treatment sessions, and one other patient reported muscle strain. No adverse events were reported in the PA group.

### 3.2. Adherence, Motivation, and Capability Regarding the Treatment Plan

Table 2 describes the reporting of adherence, motivation, and capability regarding the treatment plan at 6 months. In most cases a “high or very high grade” was reported by patients and practitioners. However, the NB group had a statistically significantly larger proportion of patients reporting a very high grade of adherence to the treatment plan compared to the PA group (*p* = 0.014). Furthermore, the NB group had a statistically significantly larger proportion of patients reporting a “very high grade” of motivation regarding the treatment plan compared to the PA group (*p* = 0.002). The NB group also had statistically significantly smaller proportion of patients reporting a lower grade of motivation regarding the treatment plan compared to the SSE and PA groups (*p* = 0.002).The strength of patient–HCP agreement in reported adherence to the treatment plan at 6 months was fair (K = 0.319). Patient–HCP proportional agreement on treatment adherence most frequently occurred within the “high grade–sure” (*n* = 23/115), and “very high grade–very sure” (*n* = 22/115) responses, but also had the largest mismatch between “high grade–very sure” (*n* = 40/115) responses.

Logistic regression highlighted that 37% of the variation in adherence at 6 months was explained by motivation and capability (R^2^ = 0.37). Patients with high self-rated motivation had 3.6 times better odds of higher adherence to the treatment plan (OR 3.58, *p* = 0.043). Likewise, patients with high self-rated capability had 14.2 times better odds of higher adherence to the treatment plan (OR = 14.24, *p* = 0.004).

**Table 2 jcm-10-04967-t002:** Adherence, motivation, and capability regarding the treatment plan.

Patient and Practitioner Reporting Related to Performance of the Treatment Plan				
	NB	SSE	PA	
	N (%)	N (%)	N (%)	*p*-Value
Patient-reported adherence to the treatment plan				**0.014**
Very high grade	**14 (37) ^a^**	7 (18)	**4 (10) ^b^**	
High grade	22 (58)	21 (54)	28 (74)	
Low grade	2 (5)	9 (23)	6 (16)	
Not at all	0 (0)	2 (5)	0 (0)	
Patient-reported motivation regarding the treatment plan				**0.002**
Very high grade	**22 (56) ^a^**	12 (30)	**6 (16) ^b^**	
High grade	14 (36)	14 (35)	19 (50)	
Low grade	**2 (5) ^b^**	**11 (27) ^a^**	**11 (29) ^a^**	
Not at all	1 (3)	3 (8)	2 (5)	
Patient-reported capability to adhere to the treatment plan				0.076
Very high grade	21(54)	15 (38)	14 (37)	
High grade	17 (44)	22 (55)	18 (47)	
Low grade	1 (2)	1 (2)	6 (16)	
Not at all	0 (0)	2 (5)	0 (0)	
Practitioner-reported patient adherence to the treatment plan				0.129
Very sure	28 (67)	18 (44)	27 (64)	
Sure	11 (26)	10 (24)	10 (24)	
Unsure	2 (5)	8 (20)	3 (7)	
Not at all	1 (2)	5 (12)	2 (5)	
Patient–practitioner concordance in reporting of adherence to the treatment plan				
	Practitioner-reported patient adherence to the treatment plan			
	Very sure	Sure	Unsure	Not at all
Patient-reported adherence to the treatment plan				
Very high grade	22	2	1	0
High grade	40	23	8	0
Low grade	4	5	3	5
Not at all	0	0	0	2
	Linear weighted Kappa = 0.319			

N: number of patients; NB: hypercorrective Boston brace night shift; SSE: scoliosis-specific exercise; PA: adequate self-mediated physical activity. Bold = statistical significance at *p* < 0.05, where row proportion ^a^ > ^b^.

### 3.3. Physical Activity Level

Results of the IPAQ-SF are displayed in Table 3. Regarding change from baseline to 6 months within groups, the NB group showed a small, statistically significant decrease in sitting time of 79 min/day (*p* = 0.031, *d* = 0.45). The SSE group had a statistically significant medium-sized increase in walking time of 20 min/day (*p* = 0.002, *d* = 0.51). The PA group had a small, statistically significant increase in moderate-intensity activity of 23 min/day (*p* = 0.009, *d* = 0.49), as well as an increase in walking of 15 min/day (*p* = 0.033, *d* = 0.35). Furthermore, the SSE and PA groups had statistically significant medium-sized increases of 1499 MET-min/week (*p* = 0.001, *d* = 0.59) and 1378 MET-min/week (*p* = < 0.001, *d* = 0.73), respectively.

Regarding between group differences in changes from baseline to 6 months, there was a statistically significant medium-sized main effect (F = 5.7, *p* = 0.004, η_p^2^_ = 0.10) for moderate-intensity activity. This favoured the SSE group, with 37 min/day more than the NB group. Similarly, there was a statistically significant medium-sized main effect (F = 6.8; *p* = 0.002, η_p^2^_ = 0.11) for walking, favouring the SSE and PA groups, with 27 min/day and 23 min/day more, respectively, compared to the NB group. The same pattern was displayed in MET-min/week, with a large, statistically significant main effect (F = 8.3, *p* = < 0.001, η_p^2^_ = 0.14) favouring the SSE and PA groups, with 1880 and 1482 more MET-min/week, respectively, compared to the NB group.

**Table 3 jcm-10-04967-t003:** Physical activity levels for the IPAQ-SF in min/day and MET-min/week at baseline, within group change and between group differences in change between baseline to 6 months.

	Baseline	Within Group Changes from Baseline to 6 Months			Between Group Differences in Changes from Baseline to 6 Months	
IPAQ-SF min/day	Mean (SD)	Mean Change (SD)	*p*-Value	Cohen´s *d*	Main Effects F; *p*-Value; η_p^2^_	Pairwise Comparison Mean difference (95% CI) *p*-Value
		Positive change = better outcome				Favoured group ↑
Vigorous					0.2; 0.844; 0.03	NA
NB	45 (54)	5 (49)	0.565			
SSE	47 (53)	9 (65)	0.392			
PA	40 (47)	9 (43)	0.228			
Moderate					5.7; **0.004**; 0.10	
NB	47 (60)	−10 (60)	0.318			NB-SSE ↑ = −37 (−64 to −10); **0.004**
SSE	65 (54)	11 (85)	0.417			SSE-PA= 10 (−17 to 38); >0.999
PA	39 (45)	23 (51)	**0.009**	0.49		NB-PA = −26 (−54 to 1); 0.066
Walking					6.8; **0.002**; 0.11	
NB	50 (46)	−14 (47)	0.073			NB-SSE ↑ = −27 (−46 to −8); **0.002**
SSE	38 (37)	20 (39)	**0.002**	0.51		SSE-PA= 4 (−15 to 23); >0.999
PA	39 (47)	15 (43)	**0.033**	0.35		NB-PA ↑ = −23 (−42 to −4); **0.014**
		Negative change = positive outcome				
Sitting					1.8; 0.167; 0.04	NA
NB	458 (199)	−79 (180)	**0.031**	0.45		
SSE	385 (184)	17 (134)	0.526			
PA	415 (153)	−9 (126)	0.693			
IPAQ-SF MET-min/week		Positive change = better outcome				Favoured group ↑
					8.3; < **0.001**; 0.14	
NB	2751 (3522)	−411 (2508)	0.332			NB-SSE ↑ = −1880 (−3059 to −701); **0.001**
SSE	2693 (2098)	1499 (2525)	**0.001**	0.59		SSE-PA= 398 (−786 to 1582); >0.999
PA	2139 (1765)	1378 (2216)	**0.001**	0.73		NB-PA ↑= −1482 (−2667 to −297); **0.009**

IPAQ-SF: International Physical Activity Questionnaire short form; MET: metabolic equivalent; NB: hypercorrective Boston brace night shift; SSE: scoliosis-specific exercise; PA: adequate self-mediated physical activity. ↑ = Favoured group marked with an arrow, Bold = significant at *p* < 0.05; Cohen’s *d* estimate = effect size; η_p^2^_ = partial eta squared. NA = not applicable.

### 3.4. Perception of Spinal Appearance

There was a small, statistically significant mean decrease in pSAQ from baseline to 6 months for the SSE group (0.9 (±2.9), *p* = 0.049, *d* = 0.34). No between group differences in change from baseline to 6 months were displayed regarding pSAQ (Table 4).

### 3.5. Angle of Trunk Rotation and Cobb Angle

ATR showed a small, statistically significant within group mean change, increasing from baseline to 6 months for the SSE (1.2 (±2.4), *p* = 0.004) and PA groups (1.0 (±3.0), *p* = 0.029). A significant between group difference in change from baseline to 6 months was displayed favouring the NB group compared to the PA group (−1.5 (CI −3.0 to −0.1), *p* = 0.037), with a medium-sized main effect (F = 4.1, *p*-value = 0.019, η_p^2^_ = 0.07) (Table 4).

Cobb angle showed a small, statistically significant within group mean change, increasing from baseline to 6 months for the NB group (2.3 (±4.3), *p* = 0.001). Likewise, there was a medium-sized within group mean change increasing for the SSE (3.7 (±7.0), *p* = 0.001) and PA groups (3.7 (±6.3), *p* = < 0.001). No between group differences in change from baseline to 6 months were displayed regarding Cobb angle (Table 4). The interrater reliability of Cobb angle measurements performed by two blinded assessors at baseline was ICC 2,1 = 0.817, and at 6 months ICC 2,1 = 0.888.

**Table 4 jcm-10-04967-t004:** Spinal appearance, angle of trunk rotation, and Cobb angles at baseline, within group changes from baseline to 6 months, and between group differences in changes from baseline to 6 months.

	Baseline	Within Group Changes from Baseline to 6 Months			Between Group Differences in Changes between Baseline and 6 Months	
	Mean (SD)	Mean Change (SD)	*p*-Value	Cohen´s *d*	Main Effects F; *p*-Value; η_p^2^_	Pairwise Comparison Mean Difference (95% CI) *p*-Value
	Positive change= worse outcome					Favoured group ↑
pSAQ general (7–35)					0.7; 0.485; 0.01	NA
NB	11.7 (3.5)	0.1 (2.9)	0.784			
SSE	11.5 (3.1)	0.9 (2.9)	**0.049**	0.32		
PA	11.7 (2.8)	0.6 (3.2)	0.261			
Angle of trunk rotation (degrees)					4.1; **0.019**; 0.07	
NB	11.8 (2.7)	−0.6 (3.0)	0.172			NB-SSE= −1.5 (−3.1 to <0.1); 0.051
SSE	10.5 (3.0)	1.2 (2.4)	**0.004**	0.34		SSE-PA= −0.2 (−1.5 to 1.5); >0.999
PA	11.1 (2.8)	1.0 (3.0)	**0.029**	0.34		NB↑-PA= −1.5 (−3.0 to −0.1); **0.037**
Cobb angle					1.1; 0.332; 0.02	NA
NB	32 (5.6)	2.3 (4.3)	**0.001**	0.34		
SSE	31 (4.7)	3.7 (7.0)	**0.001**	0.50		
PA	31 (5.6)	3.7 (6.3)	**<0.001**	0.50		

pSAQ: pictorial Spinal Appearance Questionnaire; NB: hypercorrective Boston brace night shift; SSE: scoliosis-specific exercise; PA: adequate self-mediated physical activity. ↑ = Favoured group marked with an arrow, Bold = significant at *p* < 0.05; Cohen’s *d* estimate = effect size; η_p^2^_ = partial eta squared.

### 3.6. Health-Related Quality of Life

Results for the SRS-22r are displayed in Table 5. Regarding changes from baseline to 6 months within groups, the NB group had a small decrease in the function domain (−0.1 (±0.3), *p* = 0.018, *d* = 0.36), the PA group had a small decrease in the mental health domain (−0.2 (±0.5), *p* = 0.009, *d* = 0.33), and the SSE group had a small decrease in the SRS-22r subscore (−0.1 (±0.3), *p* = 0.039, *d* = 0.22). No between group differences in change from baseline to 6 months were displayed regarding SRS−22r domains or subscores. At 6 months, the SRS-22r satisfaction domain and total score displayed no significant differences between groups.

For EQ-VAS global health state, there were no statistically significant within group changes or between group differences in changes between baseline and 6 months (Table 5). The EQ-5D-Y dimensions and Paretian Classification of Health Change showed no statistically significant differences between groups at 6 months (Table 6).

**Table 5 jcm-10-04967-t005:** Health-related quality of life according to the SRS-22r and EQ-VAS global health scores at baseline, within group changes from baseline to 6 months, and between group differences in changes from baseline to 6 months.

	Baseline	Within Group Changes from Baseline to 6 Months			Between Group Differences in Changes between Baseline and 6 Months	
	Mean (SD)	Mean Change (SD)	*p*-Value	Cohen´s *d*	Main Effects F; *p*-Value; η_p^2^_	Pairwise Comparison Mean Difference (95% CI) *p*-Value
		Negative Change = Worse Outcome				
SRS-22r Function (1–5)					1.2; 0.298; 0.02	NA
NB	4.8 (0.2)	−0.1 (0.3)	**0.018**	0.36		
SSE	4.7 (0.3)	−0.1 (0.4)	0.066			
PA	4.7 (0.3)	<−0.1 (0.3)	0.925			
SRS-22r Pain (1–5)					0.4; 0.658; 0.01	NA
NB	4.7 (0.6)	<−0.1 (0.4)	0.731			
SSE	4.6 (0.6)	<−0.1 (0.4)	0.486			
PA	4.7 (0.6)	<−0.1 (0.4)	0.943			
SRS-22r Self-image (1–5)					1.8; 0.170; 0.03	NA
NB	4.1 (0.7)	0.1 (0.5)	0.367			
SSE	4.2 (0.6)	−0.1 (0.5)	0.101			
PA	4.3 (0.5)	−0.1 (0.5)	0.200			
SRS-22r Mental health (1–5)					0.8; 0.437; 0.02	NA
NB	4.3 (0.6)	−0.1 (0.4)	0.325			
SSE	4.3 (0.5)	−0.1 (0.6)	0.152			
PA	4.3 (0.6)	−0.2 (0.5)	**0.009**	0.33		
SRS-22r Subscore (1–5)					0.9; 0.394; 0.02	NA
NB	4.5 (0.5)	<−0.1 (0.2)	0.493			
SSE	4.5 (0.4)	−0.1 (0.3)	**0.039**	0.22		
PA	4.5 (0.3)	−0.1 (0.3)	0.057			
SRS-22r Satisfaction (1–5)	At 6 months only				2.7; 0.074; 0.05	NA
NB	3.9 (0.8)					
SSE	3.5 (0.9)					
PA	3.4 (1.1)					
SRS-22r Total Score (1–5)	At 6 months only				0.8; 0.460; 0.01	NA
NB	4.4 (0.5)					
SSE	4.3 (0.5)					
PA	4.3 (0.4)					
EQ-VAS Global health (0–100)					0.2; 0.763; 0.01	NA
NB	88.2 (11.5)	−2.6 (10.4)	0.133			
SSE	88.0 (10.0)	−3.2 (15.0)	0.197			
PA	87.6 (10.6)	−1.0 (12.2)	0.620			

SRS-22r: The Scoliosis Research Society-22r; EQ-VAS: EuroQol Visual Analogue Scale; NB: hypercorrective Boston brace night shift; SSE: scoliosis-specific exercise; PA: adequate self-mediated physical activity. Bold = significant at *p* < 0.05; Cohen’s *d* estimate = effect size; η_p^2^_ = partial eta squared. NA = not applicable.

**Table 6 jcm-10-04967-t006:** Health-related quality of life for the EQ-5D-Y dimensions, at baseline and 6 months, and changes in health state according to Paretian Classification of Health Change from baseline to 6 months.

EQ-5D-Y Dimensions	Mobility		Self-Care		Usual Activities		Pain/Discomfort		Anxiety/Depression	
	Baseline	6 Months	Baseline	6 Months	Baseline	6 Months	Baseline	6 Months	Baseline	6 Months
No problems										
NB-number (%)	44 (100)	39 (100)	44 (100)	38 (100)	42 (96)	38 (97)	34 (77)	30 (77)	28 (65)	25 (66)
SSE	43 (98)	39 (98)	44 (100)	40 (100)	41 (93)	37 (92)	33 (75)	26 (65)	31 (74)	29 (74)
PA	45 (100)	38 (100)	45 (100)	38 (100)	43 (96)	37 (97)	34 (76)	28 (74)	32 (71)	29 (76)
Moderate problems										
NB-number (%)	0 (0)	0 (0)	0 (0)	0 (0)	2 (4)	1 (3)	9 (20)	8 (20)	14 (33)	12 (32)
SSE	1 (2)	1 (2)	0 (0)	0 (0)	3 (7)	3 (8)	9 (20)	13 (32)	11 (26)	9 (23)
PA	0 (0)	0 (0)	0 (0)	0 (0)	2 (4)	1 (2)	10 (22)	9 (24)	12 (27)	8 (21)
Severe problems										
NB-number (%)	0 (0)	0 (0)	0 (0)	0 (0)	0 (0)	0 (0)	1 (2)	1 (3)	1 (2)	1 (3)
SSE	0 (0)	0 (0)	0 (0)	0 (0)	0 (0)	0 (0)	2 (4)	1 (2)	0 (0)	1 (3)
PA	0 (0)	0 (0)	0 (0)	0 (0)	0 (0)	0 (0)	1 (2)	1 (3)	1 (2)	1 (3)
Chi^2^ *p*-Value	0.662	> 0.999	NA	NA	0.898	0.617	>0.999	0.869	0.852	0.906
Paretian Classification of Health Change, baseline to 6 months	Worse ^1^		Unchanged ^2^		Better ^3^		Indeterminable ^4^		Chi ^2^ *p*-Value	
NB *n* = 38-number (%)										
SSE *n* = 37-number (%)	8 (22)		22 (60)		6 (16)		1 (3)		0.448	
PA *n* = 38-number (%)	10 (26)		18 (47)		9 (24)		1 (3)			

EQ-5D-Y: EuroQol 5-Dimensions Youth; NB: hypercorrective Boston brace night shift; SSE: scoliosis-specific exercise; PA: adequate self-mediated physical activity; NA: not applicable—the distribution is a constant in self-care. Paretian Classification of Health Change: ^1^ The health state is worse in at least one dimension, and is no better in any other dimension; ^2^ the health state is exactly the same; ^3^ the health state is better in at least one dimension and is no worse in any other dimension; ^4^ the changes in health are “mixed”—better in one dimension, but worse in another.

## 4. Discussion

The present study’s results highlight that differences exist in adherence and patient-reported outcomes after a 6-month period of either NB, SSE, or PA treatments for AIS. Overall, self-reported and HCP-reported assessments of patient adherence, motivation, and capability regarding the treatments were high, especially for the NB group (92–98%), while adherence and motivation for the PA and SSE treatments were statistically significantly lower by approximately 10–20%, respectively. Importantly, patient-reported capability to carry out the treatment plan over 6 months displayed no differences between groups. In contrast, a report from the World Health Organization showed lower proportions of adherence (50%) in long-term therapies, which further reduced if treatments were complex and self-managed [26]. This implies that the HCPs in this study successfully facilitated patient and parent understanding and physical ability to perform and self-manage the treatment plans over 6 months.

In line with the COM-B model [30] and other studies [26,53], high patient adherence had a significant positive association with patient capability and motivation. This suggests the importance of the HCP strategies in the delivery of the treatment plans. This not only serves to facilitate and maintain patient capabilities, but also to improve motivation, which may be important for long-term adherence to interventions. The COM-B model highlights the importance of providing sufficient opportunity to maintain a target behaviour. For example, the provision of extra HCP guidance if requested, the use of a training diary, and the promotion of the parent’s role in guiding and motivating the patient can be important strategies that may help to influence motivation and treatment adherence.

Within the NB group, according to the IPAQ-SF, there was a decrease in time spent sitting, but no change in other physical activity levels during the 6-month period. In contrast, both the PA and SSE groups had a significant increase in walking, while moderate-intensity physical activity levels significantly increased in the PA group. This contributed to a significantly higher improvement in MET-min/week of physical activity in the PA and SSE groups compared to the NB group. Group means at baseline for IPAQ-SF levels, however, were already in line with recommendations for physical activity from the Public Health Ministry of Sweden [54], which are in line with the WHO´s recommendations [31]. Instead of increasing physical activity levels, one can speculate that the NB group was above all motivated to adhere to night-time bracing. This could be a potential factor influencing the higher patient ratings of adherence and motivation with the NB treatment plan [55]. George et al. [53] pointed out that adherence is rarely, if ever, an all-or-nothing phenomenon; more likely, some recommendations in a treatment plan are followed, while others receive less focus from the patient.

The patients in our sample had mean Cobb angle curves of 31 (±5.3) degrees at baseline, but perceived minimal spinal shape asymmetry according to the pSAQ [6]. The SAQ has been shown to be more sensitive and responsive to change than textual scales such as the SRS-22r self-image domain [28]. The SSE and PA groups had a small increase in ATR and a medium increase in Cobb angle, while small increases in pSAQ for the SSE group and Cobb angle for the NB group were seen at 6 months. However, the SRS-22r self-image domain displayed no differences during the study period. ATR displayed a medium-sized between group difference in changes from baseline to 6 months, favouring the NB group compared to the PA group. The observed changes in ATR and Cobb angle measurements in this study, however, were well below what can be considered clinically relevant, since both the scoliometer and digital Cobb angle measures have up to approximately 6 degrees standard error of measurement [56,57,58,59]. Schreiber et al. [60] found better general SAQ scores in AIS patients after 6 months of full-time bracing in combination with SSE compared to controls, in contrast to a previous study that showed more distorted perceived appearance in braced patients compared to non-braced patients [11]. It is possible that more intensive bracing [10] is required for patients to perceive changes in spinal appearance. However, when considering the risk of overtreatment, one may consider NB as a viable first step requiring lower dosage.

The SRS-22r satisfaction domain showed no between group differences despite the NB group reporting a significant decrease in the SRS-22r function domain over time. A previous study suggested that full-time brace wear can negatively impact different aspects of quality of life in 72% of the study population [12]. Conversely, another study found no decline in health in braced individuals compared to observation only [61]. The minimal important change for the SRS-22r function domain in moderate AIS has been reported to be 0.60 [62], which implies that the negative change in our study does not affect the patient in everyday activities.

The PA group showed a small, statistically significant decrease in the SRSr-22r mental health domain over the 6-month period. However, there were no statistically significant between group differences, and the PA group change was less than the minimal important change for the SRS-22r mental health domain in moderate AIS, which has been reported to be 0.55 [62]. The SSE group showed a small, statistically significant decrease in the SRSr-22r subscore over the 6-month period; this is in contrast to the findings of Monticone et al. [63], who evaluated the effect of SSE vs. traditional exercises in mild AIS, and found increased HRQoL favouring the SSE group. The SRS-22r subscores for patients in the present study, however, were slightly below normative values of 4.7 at baseline [64], but higher than previously reported values for braced, surgically treated, or untreated adults with AIS [7,65].

The EQ-VAS global health score in the present study population was high, and similar to normative data [46]. According to the EQ-5D-Y domain scores in the present study, patients very rarely reported problems with mobility, self-care, or usual activities, and none that were severe. When considering the anxiety/depression and pain/discomfort domains, severe problems were very rare, but 20–33% of the patients reported moderate problems at either baseline or 6 months, with no statistically significant between group differences in change over the 6 months. These proportions were similar to literature reporting Swedish general population normative EQ-5D-Y data for 12 and 16 year olds [46], but a little higher than normative EQ-5D scores for ages ≤ 19 [64]. Similar occurrence of pain has been reported with SRS-22r in a study of patients with AIS [4], but as high as 46% in full-time brace-treated populations evaluated with the Brace Questionnaire [12]. Anxiety and depression can be considered to be risk factors for chronic back pain [4]. This suggests the importance of helping patients showing signs of anxiety/depression and pain/discomfort to manage such problems throughout the treatment plan.

This RCT data collection was successfully carried out within the context of the Swedish public health care system. The study is therefore representative of the Swedish context, and generalizable to similar health care systems. The study’s rigorous design, adequate sample size, patient adherence to treatment plan, and good retention in prospectively collected data provide lower risk of bias to findings compared to previous studies in the field. Another strength was the use of patient-reported outcome and experience measurements to capture the patient perspective regarding the three prescribed treatments. SSE in the current study covers similar concepts and methods to those described in previous literature on scoliosis-specific treatments [8,33]. However, the current study focused on patient self-management, with three supervised sessions as a first-step intervention during the first three months, and additional sessions if needed before the 6-month follow-up.

Key limitations of this study are that the evaluations of adherence, motivation, and capability are based on the entire treatment plan as a whole, and do not distinguish between the included parts of the treatment. Lower adherence toward one of the components in the treatment plan cannot be ruled out. This study only comprises subjective means of measuring adherence, including patient-administered questionnaires and HCP reports. Measuring adherence is complex, and overestimation [66] or underestimation among patients and HCPs is possible. Review of diaries at follow-up every 6 months was an initial strategy to maintain and report adherence as accurately as possible, but patient compliance with the use of a diary varied, with some patients filling their diaries in as an afterthought in conjunction with revisit. The current study was planned in 2011–2012, when the availability of objective and patient-friendly monitoring of adherence was in progress, as well as with telehealth to enhance adherence. Advances in new technologies and measurement methods enable objective measures of time spent in brace [10], along with pedometers and accelerometer-based devices to register physical activity, inactivity, and sleep [67]. However, objective measurements can be costly and technically demanding [26]. The current study had missing values ranging from 1.5 to 31%, where the IPAQ-SF sitting score had the highest number of missing values—possibly due to difficulty in reporting, because sitting occurs continually throughout the day. A mix of self-reported and reasonable objective measures could be the best way to capture adherence behaviour to a treatment plan [26], and to get a broad picture of health aspects in AIS [68].

## 5. Conclusions

After a 6-month intervention period, self-reported and HCP-reported patient adherence were high in all groups. Patients also reported motivation and capability to carry out and perform the treatment as high, especially in the NB group. Patients in exclusively active intervention groups increased their physical activity without other clinically relevant differences between groups on other clinical measures or patient-reported outcomes. The results suggest that prescribed interventions are viable first-step options during the first 6 months.

## Figures and Tables

**Figure 1 jcm-10-04967-f001:**
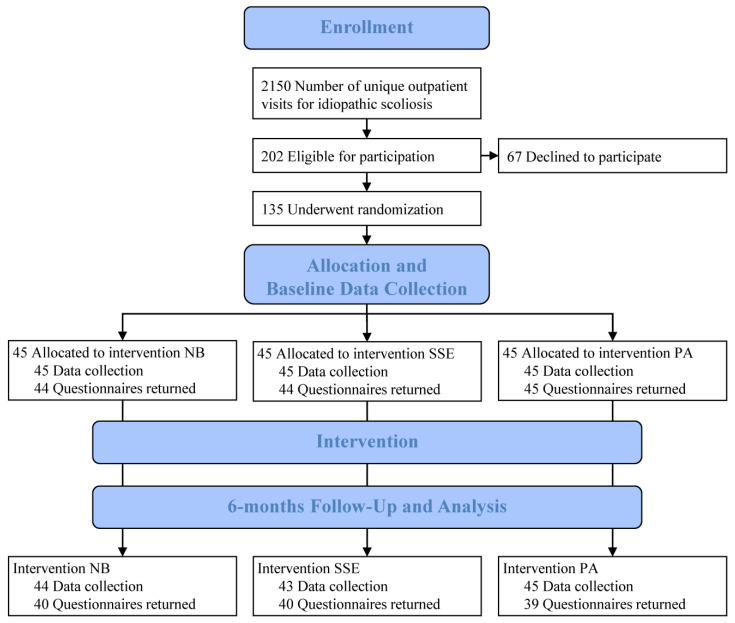
CONSORT flow diagram of the participants with AIS in the study.

## Data Availability

The data that support the findings of this study are available from the CONTRAIS research group, and are thus not publicly available. The data are, however, available from the authors upon reasonable request.

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
