# Peer review of "Six-Month Results on Treatment Adherence, Physical Activity, Spinal Appearance, Spinal Deformity, and Quality of Life in an Ongoing Randomised Trial on Conservative Treatment for Adolescent Idiopathic Scoliosis (CONTRAIS)"

_jcm, 2021, doi:10.3390/jcm10214967_

Round 1

Reviewer 1 Report

This is a promising RCT that seems to be currently going on about different conservative treatment options in patients with AIS. The study design has been previously published in 2013 (https://doi.org/10.1186/1471-2474-14-261) and i am wondering about the reason of that time gap between the publication of the protocol and this manuscript with partial results.

Additionally the main outcome of the study protocol (treatment failure) is not presented here due to the need of measuring Cobb angle changes in 2 consecutive x-rays 6 and 12 months after baseline. I am not really sure about what had happened during this 8 years because it is enough time for, at least, have the 2 follow-up measures of Cobb angle in the vast majority of patients. Adittionally, in methods it is pointed that in october 2018 the recruitment of patients finished so in october of 2019 all data regarding Cobb angle might have been taken and presented here as the main outcome of the study.

Beyond that doubt, Introduction is clearly written, well referenced and useful to understand the aim of this research. It is worth being greatful to the authors for introducing accordingly. 

In Methods, you have explained how measurements of Cobb angle were taken but results from 6 months follow-up are not presented in the study. I missed a lot this changes between Cobb angle at the baseline and 6 months later. You have promised a new paper with the main outcome of the study protocol published in 2013 at it seems to be like a TV series by chapters. The only reason that could explain main outcome absence in this paper is that 12 months follow-up has not been made yet. But it seems that it has been done according to the dates you give here.

This is the paragraph i am talking about: "In the ongoing RCT, the primary outcome is failure of treatment defined as an increase of Cobb angle of more than 6 degrees, seen on two consecutive x-rays after 6 months in relation to the baseline x-ray [27]. Our primary outcome will be reported in a future publication when the RCT is completed". I would like to have some explanations about how is it possible to recruit the last patient in october 2018 and to still be ongoing after more than 24 months when 12 months are enough to report Cobb angle changes according to your explanations. 

Table 1 is pretending to show no differences between the 3 groups but only mean and SD is reported. It must be completed with the appropiate statistical analys (I have suggested one in comments on the PDF). The rest of results are well presented in many tables. I know that maybe figures are not suitable for this presentation and that the high variability of analysis done need a lot of tables. It difficults a bit reading the presence of those high number of tables but i understand that this is the best way found by the authors. A hard work is clearly evidenced here and it improves the quality of the study and the wheigh of the conclusions of the authors. So congrats for your big effort in presenting the results.

Reviewer 2 Report

The manuscript “6-month results on treatment adherence, physical activity, spinal appearance, spinal deformity and quality of life in an ongoing randomised trial on CONservative TReatment for Adolescent Idiopathic Scoliosis (CONTRAIS)” by Marlene Dufvenberg ,  Elias Diarbakerli, Anastasios Charalampidis, Birgitta Öberg, Hans Tropp, Anna Aspberg Ahl, Hans Möller, Paul Gerdhem, and Allan Abbott is an article that aimed to explore patient adherence to treatment and secondary outcomes during the first 6 months for physical activity, spinal appearance, spinal deformity and quality of life within and between NB, SSE, and PA based treatments for AIS in an ongoing RCT.

Interesting results.

Below are my comments and remarks regarding the article:

1. Adherence, the definition in the introduction seems unnecessary
2. The duration of the study should be significantly extended for better assessment of the effects of treatment and compliance
